# Testing of a Program to Automatically Analyze Students’ Concept Maps

**DOI:** 10.3390/pharmacy8040209

**Published:** 2020-11-07

**Authors:** Robert Hubal, Laura Bobbitt, Sarah Garfinkle, Suzanne C. Harris, Brandon D. Powell, Madison S. Oxley, Heidi N. Anksorus, Kevin Y. Chen

**Affiliations:** 1Division of Practice Advancement and Clinical Education, UNC Eshelman School of Pharmacy, Chapel Hill, NC 27599, USA; laurajbobbitt@gmail.com (L.B.); sgarfinkle4@gmail.com (S.G.); suzanne_harris@unc.edu (S.C.H.); hanksoru@email.unc.edu (H.N.A.); 2Department of Pharmacy, University of North Carolina Medical Center, Chapel Hill, NC 27514, USA; brandon.powell@unchealth.unc.edu (B.D.P.); Kevin.Chen2@unchealth.unc.edu (K.Y.C.); 3Department of Pharmacy, Virginia Commonwealth University Health System, Richmond, VA 23298, USA; madison.oxley@vcuhealth.org

**Keywords:** concept maps, software analysis, student assessment

## Abstract

Concept maps are graphical representations of how various concepts relate to one another. Assessment of concept maps developed by students in the pharmacy curriculum helps to evaluate student understanding of course material. However, providing feedback on concept maps can be time-consuming and often requires the grader to be a content expert. The purpose of this study was to develop and validate a software program to provide students with feedback on their concept map performance. Student maps for four different disease states were compared against expert concept maps. The analysis of the program compared favorably to a manual assessment of student maps for the maps’ complexity and content but did not correlate for their organization. The value of using a software program to quickly and efficiently analyze concept maps is discussed.

## 1. Introduction

Metacognition, or “thinking about your thinking”, is an essential component of developing study skills, self-directed learning, and critical thinking. As such, metacognitive ability is important for both student pharmacists’ success in the classroom, as well as their continued success as independent clinicians. High performing students have demonstrated more accurate self-assessment of their ability as compared to low performing students, and are better able to identify incorrect items on an exam [1,2]. Additionally, when asking students about their confidence and willingness to ask for help on specific topics, students may show overconfidence in topics they are less familiar with, and be less willing to ask for help with these same topics [3]. This finding highlights the need to identify and implement strategies to enhance student metacognitive skills within their curriculum. Various strategies can be employed to enhance student metacognition in the classroom including exam reviews, reflections, and adding judgment of understanding [4]; the focus of this study was another strategy: the use of concept maps.

Concept maps are graphical representations of how various concepts are related to one another. They are used as an educational technique to help improve students’ understanding of course material, integrate didactic and experiential knowledge, and encourage higher-order learning [5]. The organization and complexity of concept maps can help identify different types of knowledge that the students possess, and help to inform tutoring [6,7]. Administration of a concept map activity where nurses were asked to map out a patient’s primary health need, key assessment findings, diagnosis, and interventions helped to increase critical thinking skills using a nursing validated assessment [8]. The technique has also been useful for assessing the effectiveness of education sessions used to develop critical thinking skills in new pediatric medical residents [9] and the value of structured feedback on physiotherapy students’ conceptual knowledge [10]. Although concept mapping has been shown to be effective in fostering and evaluating critical thinking skills in some fields, such as nursing and physician education, the broad utility of concept maps for the development of metacognition remains under-studied.

The use of concept mapping in the pharmacy curriculum is becoming more prevalent in recent years as shifts to accreditation standards encourage more engaged and active learning. Concept maps are currently used to facilitate the learning of disease state knowledge such as pain and cardiovascular diseases, as well as an understanding of the Pharmacists Patient Care Process [11,12,13]. Furthermore, concept maps have also been used as an assessment device in pharmacy classrooms [14]. By using concept maps as an assessment device, two strategies of enhancing metacognitive skills—reflections and adding judgment of understanding—can be applied to these maps to determine their effect on improving student self-assessment, an essential component of metacognitive knowledge. In the authors’ school of pharmacy, concept maps are being used to help guide self-directed learning of foundational material on therapeutic topics—pre-class work-ups that students build on with cases and expert debriefs. However, they can also serve as a type of formative assessment. When paired with feedback, and compared to expert maps, concept maps offer the potential to allow students to self-assess or be assessed on their understanding of therapeutic topics.

Importantly, these efforts take time and resources. To provide feedback on concept map performance and enable students to reflect on their performance requires several steps: (1) developing an expert concept map (called a ‘key’), (2) having students perform the concept mapping activity, (3) comparing each student map against the key using common measures of network similarity (such as overlap in the number and labeling of concepts, overlap in the types of relationships that are shown, and the maps’ overall organization and complexity), and (4) reporting salient similarities and differences between the student map and key that might benefit students. However, the effort involved might preclude the use or affect the value of concept maps in certain educational settings. In this study, we aimed to investigate the validity of results produced by a software program against expert-rated grades and to assess the program’s ability to automate part of the feedback process and thus lessen the effort involved.

## 2. Materials and Methods

### 2.1. Data Collection

Second-year students in a core curriculum pharmacotherapy course, all enrolled in the Pharm.D. program at an accredited school of pharmacy in North Carolina, concept mapped a pseudorandomized set of disease states over the course of a semester [15]. They did so in fixed groups of up to eight students; there were 16 groups in total. (Hereafter these are termed student maps, though, in reality, they were student group maps.) At the beginning of the semester, a faculty member modeled for the students how to make a concept map of a disease state utilizing six core domains (pathophysiology/etiology, diagnostics, signs and symptoms, goals of care, treatment, and monitoring/follow up); these domains are consistent with the literature on patient-centered care [16]. The domains serve as an organizational structure, thereby reducing variability between student maps and an expert key. During the course, whenever there was a concept map activity, it was completed prior to the first lecture on the given disease state to establish baseline knowledge. After the activity, students were prepared to engage in cases and higher-order learning by self-comparing their maps against the expert key.

In their assignments, student groups were permitted to create a concept map electronically or on a whiteboard. All concept maps were turned in for a completion grade. Groups that chose to create their map on a whiteboard took pictures of the map to turn in. Eight concept maps (out of 16 groups) were randomly selected for assessment for each of four disease states—asthma, chronic kidney disease (CKD), heart failure (HF), and rheumatoid arthritis (RA)—for which there was a concept mapping assignment. All analyses were conducted on anonymized maps after course completion; the results of this study did not affect student course grades. The study was approved by the Institutional Review Board of the authors’ university.

Expert keys for each of the four disease states were developed by two postgraduate first year (PGY1) resident pharmacists involved with the study, with input and approval by content experts who teach the lectures in pharmacotherapy.

### 2.2. Data Transcription

The selected student maps were transcribed by two fourth-year (PY4) student pharmacists using an off-the-shelf mind-mapping tool (www.freeplane.org). These transcribers corrected spelling errors, aligned capitalization to the key, and converted synonyms and abbreviations used in the disease state concept maps to match the corresponding key. The central node for any given map was the associated disease state (see Figure 1). From that node in the expert key were always six connections, referred to here as links, one to each domain (pathophysiology/etiology, diagnostics, signs and symptoms, goals of care, treatment, and monitoring/follow up). Nodes could differ between student maps and expert keys, as could links between nodes. For the most part, a link connected a node to a parent or child node, but on occasion, a link crossed to another path in the network; these were labeled as ‘crosslinks’. Student groups were encouraged to add crosslinks when appropriate, essentially demonstrating critical thinking in showing relationships across the hierarchy of domains (e.g., from diagnostics to treatment). When transcribing crosslinks, if a student group’s crosslink was more descriptive when compared to the expert key, then the crosslink was transcribed to replicate the key. In other words, if the student groups’ crosslink linked to a child (or grandchild) node to that of the key, and thus was more specific than the key, then it was assumed that the students appropriately understood how to link the concepts.

### 2.3. Organization and Complexity

A given concept map can be more or less organized and complex than a comparison map (e.g., an expert key). In a simple, organized concept map, the number of nodes is roughly the same as the number of nodes in the comparison map, the number of links is on the order of the number of nodes, and there are few crosslinks. In a complex or disorderly map, the number of nodes differs from the number of nodes in the comparison map, and the number of links and/or crosslinks differs from the number of nodes. A program to efficiently compute network characteristics—count nodes and links and determine organization and complexity—could be useful for students in providing specifics in how their concept maps resemble or differ from the expert key.

A Python program was written to transform the file format used by the mind-mapping tool to extensible markup language (XML). From there, a standard XML diff command generated differences between the student map and the expert key, and custom code was written to eliminate formatting, clean the diff, and produce interpretable results (Table 1). Missed nodes, extra nodes, same links, different links, and crosslinks were all analyzed by the software program, as were measures of complexity of the concept map and its organization. Complexity value was calculated as the total number of elements (nodes, links, and crosslinks) in the given student map against the total number of elements in the expert key. A measure close to 1 suggests a match to the key; a measure lower than 1 suggests a more simple student map, while a measure greater than 1 suggests a more complex student map, compared to the expert key. The organizational value was calculated as an average of the individual percentages of student map nodes, links, and crosslinks matching relative to the expert key. A measure close to 1 suggests a match to the key; a measure lower than 1 suggests a more organized student map, while a measure greater than 1 suggests a more disorderly student map, compared to the expert key. In an iterative process, the transcribers checked the program’s output for accuracy and the programming team addressed and resolved any errors.

### 2.4. Data Comparison

To validate the software program, its results were compared to manual expert analyses of the concept maps. The focus was primarily on content and criterion-related validity [17,18], in essence, checking if the program could address how domain-related concepts are structured and if it produced results convergent with experts. One set of analyses involved comparing the organization and complexity of the student maps against expert keys, as described above. Another set of analyses involved comparing these network characteristics not against other maps but against grades. Experts were the same PGY1 residents who developed the keys, along with a school of pharmacy faculty member. The experts worked together, using a previously published rubric, to ensure consistent grading, performed as a separate part of the study [15]. The rubric [13] was used to grade concept maps, based on concepts identified, on a scale of good, better, or best. This scale was translated for purposes here to a scale of 1 to 3.

## 3. Results

### 3.1. Network Characteristics

By having the PY4 students transcribe student group concept maps using an off-the-shelf mind-mapping tool, it was possible to determine network characteristics of the maps, including their number of nodes (concepts) and links (relationships between concepts) and measures of network structure such as complexity and level of organization. The PY4 students gauged the transcription took on average five minutes per concept map but ranged up to a half-hour for sizeable maps. The results are shown in Table 2.

The size of the networks was similar for three of the disease states (CKD, RA, and asthma), but that of HF was larger. This was true for both the expert key and across student groups. For the number of nodes, student maps were on average about the same as the expert key. For the number of links, student maps were again on average the same as the expert key though there was a slightly lower (non-significant) number for students for rheumatoid arthritis.

Of greater interest here were the network calculations performed on the concept maps. Over the course of the semester, student group concept maps trended toward less organization but more complexity, compared to the expert key, in line with other studies finding that knowledge patterns show an inverted-U as students progress from novices to intermediates to experts [19,20]. A correlation was run between student group concept map organization and complexity and the expert-assigned grades of good, better, or best, to determine if the calculated complexity and organization metrics correlated with the expert-assigned values. On average, the correlation value between the calculated complexity measure and the expert-rated complexity measure was significant at 0.31 (*p* < 0.04), whereas the correlation between the calculated and rated organization measures was not significant.

### 3.2. Necessity of Transcription

As a check on the transcription effort expended by the two PY4 students, the analyses were re-run using the student groups’ original concept maps rather than the transcribed maps. That is, these maps were implemented in the off-the-shelf mind-mapping tool but not re-coded for spelling errors, synonyms, or abbreviations. At first, not aligning student maps with the expert key in this way resulted in neither correlation of complexity nor organization with the expert-rated measures reaching significance. However, the student groups’ original maps were then run through a straightforward Python routine that merely searched for and replaced key matches (e.g., ‘diagnosis’ and ‘diagnostics’ with ‘Dx’; ‘history’ with ‘Hx’; ‘signs’, ‘symptoms’, ‘clinical presentation’ with ‘Sx’; ‘hypertension’ with ‘HTN’). This routine focused on the content within the maps, not their structure. As a result, the correlation value between this calculated complexity measure and the expert-rated complexity measure was 0.34, significant again at *p* < 0.03, whereas the correlation between the calculated and rated organization measures was, as before, not significant.

One more analysis was conducted to further explore the value of the search-and-replace routine just described, when applied to student group concept maps. Additional Python code was written to assess the degree of similarity, using a calculation sometimes called ‘intersection over union’ that gauges overlap between two datasets (e.g., adapted from [21]). In this case, the calculation was done between each concept map and the associated key. The calculation was done for all three sets of maps: those transcribed by the PY4 students, and as just mentioned those that were the student groups’ original concept maps without any search-and-replace and those where search-and-replace was applied. Results from this program showed, on average, 10% overlap with the expert keys for the original maps without searching and replacing, 22% overlap for the original maps with searching and replacing, and 24% overlap for the PY4-coded maps. There were significant differences by a *t*-test (*p* < 0.01) between not searching and replacing and performing searching and replacing, and between not searching and replacing and having the PY4 students transcribe maps.

## 4. Discussion

This study showed positive early results from investigating what aspects of a concept map can be easily assessed using a software program for grading convenience. The focus was on what the program can do to assess maps, in comparison with expert grading. Certainly, there is thought that is put into manual grading that feeds into a good, better, best evaluation. However, such expert grading can be subjective, and a three-point scale leaves little to separate map attempts. The transcription by PY4 students took some effort. Yet the benefit of having transcribed student maps using an off-the-shelf mind-mapping tool is that maps could then be exported using a readily manipulable text format. The automated program developed for this research was able to capture much of the variability in expert grading associated with map complexity, though currently, it is unable to capture variability associated with maps’ organization. This finding might be due to the highly structured format of the maps—the central and all six domain nodes were equivalent a priori—that resulted in differences between students and experts primarily on more specific nodes and links.

The study showed several advantages to using a software program for assessing maps. First, the program was efficient; once the maps were coded using the mind-mapping tool, the comparison was immediate. This particular mind-mapping tool, as are others, is easy to use and could be employed by students themselves to create maps. Because most all of these mind-mapping tools represent the maps in an XML-like file format, they do away with the need for any transcription. Indeed, the search-and-replace routine was sufficient to allow for student maps to be directly compared to the expert keys, yielding results in line with the PY4-transcribed maps. Second, the program was able to generate additional data on the differences between student maps and expert keys; in particular, the numbers of same, missing, and extra nodes, links, and crosslinks were calculated. Having the means to assess specific map differences is helpful for instructors and facilitators to identify and prioritize concepts and connections that students are missing, moving, or adding. Expertise literature suggests that concept maps of those with some proficiency differ systematically from expert maps [22], allowing for systematic exploration of meaningful and informative map differences. Thus, the value of the program is it offers more information more readily in the comparison of students’ mapping with experts’ mapping.

### 4.1. Limitations

As it stands, there are some limitations to the software program’s efficacy. First, because the authors’ school does not currently support any single mind-mapping tool, PY4 students were needed to manually transcribe the student concept maps from whiteboards and other media. The advantage was the alignment of vocabulary and structure that the PY4 students’ manual effort yielded, although it turned out that just entering the student maps into the mind-mapping tool would have sufficed, with the use of the search-and-replace routine. Additionally, when there are transcribers, they must have some knowledge of the disease states to appropriately interpret what and how to transcribe. This observation carries over to the search-and-replace: The current routine only looks at fifty or so patterns that were found in these concept maps, but a truly generic program would require hundreds of patterns. These limitations are not insurmountable; the team has designs for developing additional software to allow for consistency via drop-downs and restricted entries and an online portal for adding patterns to the search-and-replace routine, plus specialized openly-available machine learning applications have been trained on millions of clinical records and thus can take into account synonyms and abbreviations [23].

Second, there was imposed organization on the concept maps; the central node and its immediate six children (the domains) were a given. The imposition has educational value but does not allow for other graph analytic calculations such as centrality and clustering [24]. Further, the equations used in this study for complexity and organization measures might be modified. For instance, they might take into account the lowered variability in the organizational structure. Similarly, they might explicitly consider if the students drilled down to more detail than is provided in the key. Third, there was no purposeful attempt in this study to assess accuracy. These concept maps are being used for possible support of students’ metacognition [15], but not for correctness, as there is typically no single correct map (the expert key represents just one vetted map) [17,25]. Fourth, the data available for this validation study was limited to a subset of disease states and student group submissions rather than individual submissions. More data will allow for the refinement of the complexity and organization algorithms.

### 4.2. Future Studies

Existing strategies to enhance metacognition currently are limited by the time it takes to implement them effectively, thereby restricting the amount of time that can be focused on developing metacognitive skills within the curriculum [26]. The software program described here, applied to maps created with nearly any off-the-shelf mind-mapping tool, can be used to efficiently grade and provide feedback to any size group of students, providing immediate results. Further research is needed to fill gaps identified in this study related to aligning the expert maps with student maps. For instance, it has been shown that intermediate learners generally draw more interconnected maps than experts ([19,22]; see also [27]), so that mere counts of missing or moved nodes or links may miss the nuance in how the student maps demonstrate evolving understanding of the content. Similarly, the expert maps included only a few crosslinks, to limit the maps’ complexity, but disease state characteristics often cross domain boundaries, so that some relationships that might have value (e.g., aligning specific diagnostics with signs and symptoms) were left out. Concept maps can thus be studied to determine how they enhance students’ understanding of the inter-relatability among etiology, diagnosis, pharmacotherapy options, and monitoring of efficacy and safety within various disease states. Future work can also examine the impact of curricular concept mapping on student application of metacognitive skills both within the classroom and during experiential learning (i.e., experiences outside the classroom).

## 5. Conclusions

Concept maps are increasingly used to help students understand to-be-learned material, and to assist in their metacognition—their knowledge of what they do and do not know, and how to address gaps. While there are readily available mind-mapping tools, none of these tools provides easy-to-use diagnostic data comparing student maps against expert maps to assist in the learning. The software program described in this study begins to address this need, by showing how it can produce complexity and organizational measures that may align with human rater assessments. The program is easy to run and available for others to adapt and test in their classrooms, as further refinement promises greater access and diagnostics for students and instructors alike.

## Figures and Tables

**Figure 1 pharmacy-08-00209-f001:**
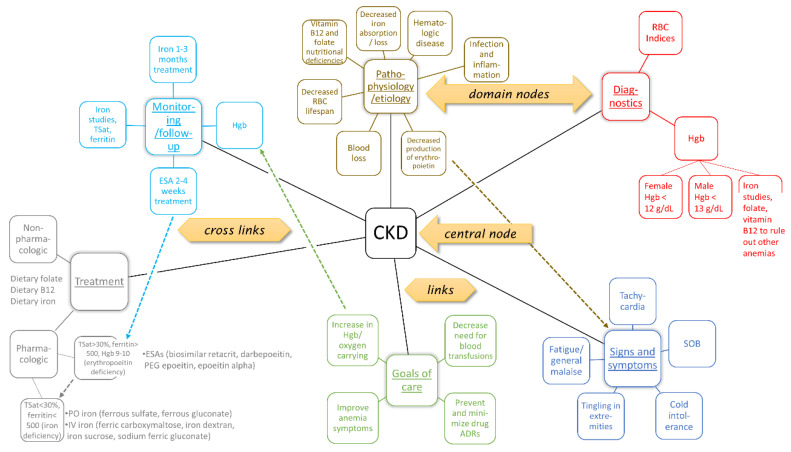
Example key for chronic kidney disease.

**Table 1 pharmacy-08-00209-t001:** A sampling of manipulated text used by a concept mapping program to describe maps. Ignored attributes do not contribute to the counts of nodes, links, and crosslinks, nor to complexity or organization measures.

Snippet of Off-The-Shelf Mind-Mapping Tool File Format	Description
<node TEXT=“CKD Key” ID=“ID_1351191738” CREATED=“1569033052754” MODIFIED=“1569035412007” STYLE=“oval”>	Definition of central node; only TEXT and ID attributes are not ignored
<node TEXT=“Pathophysiology Etiology” POSITION=“right” ID=“ID_987834383” CREATED=“1569034276739” MODIFIED=“1569034286521”>	Definition of a domain node; only TEXT and ID attributes are not ignored
<edge COLOR=“#7c0000”/>	Ignored
<node TEXT=“Decreased Iron Absorption/Loss” ID=“ID_1560551453” CREATED=“1569035171943” MODIFIED=“1569035183018”/> <node TEXT=“Hematologic Disease” ID=“ID_1679789674” CREATED=“1569035183752” MODIFIED=“1569035188168”/> ... <node TEXT=“Decreased production of erythropoietin” ID=“ID_1647921721” CREATED=“1569035232921” MODIFIED=“1569035716940”>	Definition of domain children nodes (parent domain node has not yet been closed); only TEXT and ID attributes are not ignored Note that last node is not yet closed, because it includes a crosslink.
<arrowlink SHAPE=“CUBIC_CURVE” COLOR=“#000000” WIDTH=“2” FONT_SIZE=“9” FONT_FAMILY=“SansSerif” DESTINATION=“ID_24856304” STARTARROW=“NONE” ENDARROW=“DEFAULT”/>	Definition of a crosslink (parent node—in this case, a child to the domain node—has not yet been closed); only DESTINATION attribute is not ignored
</node>	End of definition for crosslink’s parent node
</node>	End of definition for domain node
</node>	End of definition for central node

**Table 2 pharmacy-08-00209-t002:** Analyses of and comparison between student maps and expert key.

**Disease State**	**Average across N = 8 Student Maps**		**Expert Key**		**Student Maps vs. Expert Key**	
	**# Nodes**	**# Links**	**# Nodes**	**# Links**	**Organization *** **[min–max]**	**Complexity *** **[min–max]**
Chronic kidney disease	50.1 ± 11.5	36.1 ± 16.5	54	57	0.46 ± 0.11 [0.30–0.69]	0.87 ± 0.20 [0.57–1.28]
Rheumatoid arthritis	51.1 ± 17.3	40.8 ± 14.1	64	67	0.82 ± 0.31 [0.26–1.10]	0.80 ± 0.26 [0.50–1.25]
Heart failure	81.5 ± 15.7	82.5 ± 12.9	77	80	1.14 ± 0.34 [0.45–1.58]	1.07 ± 0.19 [0.79–1.26]
Asthma	61.9 ± 16.2	46.8 ± 13.4	55	57	1.50 ± 0.54 [0.48–2.27]	1.17 ± 0.30 [0.91–1.75]

* 1 = matched to key; >1 = less organized/more complex than key; <1 = more organized/less complex than key.

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
