# Peer review of "Testing of a Program to Automatically Analyze Students’ Concept Maps"

_pharmacy, 2020, doi:10.3390/pharmacy8040209_

Round 1

Reviewer 1 Report

Initially, I would like to thank you for the opportunity to review the manuscript.

It is a manuscript that describes the development and testing of a tool for assessing conceptual maps in the course of pharmacotherapy. The theme is interesting and important for the development of teaching and learning assessment methodologies, and may bring insights for the improvement of assessment strategies and facilities for the professor's work.

However, I felt some difficulty to fully understand the proposal and I explain below some issues that, in my understanding, can help the reading, understanding and evaluation of the proposal presented:

Introduction:

The text provides an interesting overview of metacognition, development and use of maps in the development of clinical reasoning and professional education. However, the subject of the study is the development and validation of a software program for the assessment of concept map learning for pharmacy students. Therefore, some basis and justification for this focus is lacking.

Methodology:

The text describing the methodology seems to me confused. It is not clear what the steps are for:

- use of concept maps by students and selection of those who entered the study sample

- software development

- processing of map content to enter the software

- comparative study of map assessment methods, to verify the validity of the software.

It became particularly complicated in the text to identify which steps were applied for the development of the software, and which are the steps for processing the contents of the students' maps that will always be necessary to use the software (transcription, coding ...). This needs to be clear to assess the usability of the proposed software in everyday pharmaceutical education. So far, I am not sure whether application of the software facilitates or complicates the work process for the assessment of student maps.

Results and discussion:

The same doubts of the methodology are reflected in the presentation of the results and discussion.

I suggest that the presentation of the methodology in subitems should be repeated in the presentation of the results. The usability of the tool needs to be better discussed.

Author Response

Initially, I would like to thank you for the opportunity to review the manuscript.

THANK YOU TO THE REVIEWER FOR THEIR IMPORTANT COMMENTS AND SUGGESTIONS.

It is a manuscript that describes the development and testing of a tool for assessing conceptual maps in the course of pharmacotherapy. The theme is interesting and important for the development of teaching and learning assessment methodologies, and may bring insights for the improvement of assessment strategies and facilities for the professor's work.

However, I felt some difficulty to fully understand the proposal and I explain below some issues that, in my understanding, can help the reading, understanding and evaluation of the proposal presented:

Introduction:

The text provides an interesting overview of metacognition, development and use of maps in the development of clinical reasoning and professional education. However, the subject of the study is the development and validation of a software program for the assessment of concept map learning for pharmacy students. Therefore, some basis and justification for this focus is lacking.

INDEED THE FOCUS OF THE STUDY IS THE SOFTWARE AND NOT METACOGNITION. METACOGNITION WAS WHAT MOTIVATED THE ORIGINAL RESEARCH BUT NOT THIS PART OF THE STUDY. THE INTRODUCTION WAS MODIFIED TO BETTER PORTRAY THIS FOCUS.

Methodology:

The text describing the methodology seems to me confused. It is not clear what the steps are for:

- use of concept maps by students and selection of those who entered the study sample

- software development

- processing of map content to enter the software

- comparative study of map assessment methods, to verify the validity of the software.

It became particularly complicated in the text to identify which steps were applied for the development of the software, and which are the steps for processing the contents of the students' maps that will always be necessary to use the software (transcription, coding ...). This needs to be clear to assess the usability of the proposed software in everyday pharmaceutical education. So far, I am not sure whether application of the software facilitates or complicates the work process for the assessment of student maps.

AS IT IS IMPORTANT THAT THE MESSAGE COME THROUGH EASILY TO THE READER, THE MANUSCRIPT HAS BEEN EDITED CAREFULLY TO MAKE THESE STEPS AND IMPLICATIONS MORE CLEAR.

Results and discussion:

The same doubts of the methodology are reflected in the presentation of the results and discussion.

I suggest that the presentation of the methodology in subitems should be repeated in the presentation of the results. The usability of the tool needs to be better discussed.

THESE SECTIONS OF THE MANUSCRIPT HAVE BEEN REVISED AGAIN TO CLARIFY.

Reviewer 2 Report

Overall, the study is novel and valuable due to the application of software to an important educational problem. The idea, itself, is worthy of publication. I understand the study methods and analysis are a small step forward in the complex development and testing of new software. However, the writing of the manuscript requires improvements for the reader to better understand the intellectual merit and implications of this study.

Overall, I recommend that the authors slow down and explain terminology. Be consistent in this terminology use. Make sure your Methods and Results section follow the same order so someone can read the Results and go back to your Methods Section to understand what was done.

Overall, the alignment between the Introduction, Methods, and Results could be improved. As currently written, if you read one section, you would expect something different in the next section. While you are working on your next revision, I recommend sitting down with a colleague who knows nothing about the study and having them read it in front of you so they may ask questions. Any questions they ask would indicate a need for greater clarity. Also, after they read each section (e.g., Intro), you could ask them “what do you expect to be in the next section?” I hope my review improves the clarity of your writing so readers have a better understanding of the value of your work.

Introduction

- [Consideration] Line 39 – Your study has no measure of metacognition but the Introduction primes the reader to think it will. As written in your methods section, your concept map activity has the purpose of “establishing baseline knowledge so that students were prepared to engage in cases and higher-order learning.” In your study, you measure the complexity and organization of students’ concept maps. However, these are not discussed in the Introduction. What is the previous literature on these constructs? Why are they important? What the gaps in calculating these measures?

- [Consideration] There is greater opportunity to set up the problem you are attempting to solve. In this case, that concept maps are an effective tool to help student organize their knowledge but that they take significant resources to provide feedback.

 - [Consideration] Line 49 – As currently written, I could not identify any reflection or self-assessment activities in your concept map activity. Consider whether you are over-applying the theory of metacognition and self-regulation and whether it aligns to your study and learning activity.

 - [Recommendation] Can you greater specify your research aims? Investigate what? The utility, feasibility and usability of the program? You have compared the results of the software with PGY1 raters. How does this match to a more specific aim?

Methods

- [Recommendation] Recommend to start section with a traditional population paragraph of what year the students were, how many cohorts, size of the cohort, degree program, country etc.

 - [Recommendation] Include data collection procedures, how students were consented/informed about the study, recruitment procedures. Was this study approved by your ethics committee?

 - [Recommendation] When I started to read the first paragraph of the Results Section, I was surprised to see these types of results. Could you set this up more in the Methods Section so the reader understands the data analysis? The methods section does do a great job of defining what nodes and links mean. However, it is less clear what aims you are meeting and why. It could be as simple as writing in the Methods Section that you calculated some descriptive characteristics of the students’ concepts maps by means of XYZ..etc.

 - [Recommendation] Similar to previous comment, can you map out chronologically your data analysis plan in the same order that is presented in the Results Section. Potentially, you might consider moving some of what is written in the Results Section to Methods.

 - [Consideration] Can you set up your reasoning for comparing the answers of the software to PGY1 raters. In this case, explain to the reader that this provides evidence of validity of the software program. You could cite similar studies to demonstrate that this is a typical process in software development? The readers would be novices in this subject area.

Results [Recommendation]

- p5. Line 138-142: You have provided a correlation before providing the raw data. What is the “complexity measure” and “organization metric” for the software? For the PGY1 raters? Please provide this data in text or table first before the correlation. Or maybe I am confused by the terms.

 - P5, line 143: What does a check on effort mean? How is the re-run different than the original analysis? It may be the sentence structure of the first few sentences that is unclear. So did you re-run everything after applying a spell checker? Or after a human looked for matches?

 - line 155: What is the “simple text-replacement program”?

- Line 159 – What is the significance of a 22% and 24% overlap? Any expert opinion on how significant those values are? Also, what does this mean? What exactly is the difference between the PGY1 coded maps and the “original maps with text replacement”? Are you saying that PGY1s marked the students maps AND that they were the ones that re-coded for spelling errors? Again, I think this would all make more sense if the Methods Section followed a clear chronological order for what calculations and procedures were conducted. I recommend to be especially clear as to who (i.e., research team, PGY1s, students) or what (i.e., software, statistical programs) is doing each calculation/procedure.

 - If you have this available - how much time did it take the human graders?

 - Table 2: Does the N=8 mean that only 8 student concept maps were evaluated?

Discussion

 - You have discussed complexity and organization but what about the accuracy of students’ maps? Was this evaluated? Will this ever be able to be assessed by the software OR will this component always need to be provided by a human rater?

 - [Consideration] Authors could choose to improve clarity by using more active voice and less passive voice sentences. Line 206 is a perfect example of this.

 - [Consideration] Consider adding a statement of whether this software is ready for use in a classroom.

- As I am still confused by some results, I cannot fully comment on the Discussion section. I am left thinking the software just provides a metric for the student concept map’s level of complexity and organization. I am unsure if it provides a % overlap with each map and the key. I am unsure if students can use the software for formative assessment – i.e., does it show students where they are lacking complexity or organization?

Author Response

Overall, the study is novel and valuable due to the application of software to an important educational problem. The idea, itself, is worthy of publication. I understand the study methods and analysis are a small step forward in the complex development and testing of new software. However, the writing of the manuscript requires improvements for the reader to better understand the intellectual merit and implications of this study.

Overall, I recommend that the authors slow down and explain terminology. Be consistent in this terminology use. Make sure your Methods and Results section follow the same order so someone can read the Results and go back to your Methods Section to understand what was done.

Overall, the alignment between the Introduction, Methods, and Results could be improved. As currently written, if you read one section, you would expect something different in the next section. While you are working on your next revision, I recommend sitting down with a colleague who knows nothing about the study and having them read it in front of you so they may ask questions. Any questions they ask would indicate a need for greater clarity. Also, after they read each section (e.g., Intro), you could ask them “what do you expect to be in the next section?” I hope my review improves the clarity of your writing so readers have a better understanding of the value of your work.

THANK YOU TO THE REVIEWER FOR THEIR COMPREHENSIVE AND INSIGHTFUL COMMENTS. PLEASE SEE BELOW FOR SPECIFIC RESPONSES.

Introduction

- [Consideration] Line 39 – Your study has no measure of metacognition but the Introduction primes the reader to think it will. As written in your methods section, your concept map activity has the purpose of “establishing baseline knowledge so that students were prepared to engage in cases and higher-order learning.” In your study, you measure the complexity and organization of students’ concept maps. However, these are not discussed in the Introduction. What is the previous literature on these constructs? Why are they important? What the gaps in calculating these measures?

UNDERSTOOD. THE FOCUS OF THE STUDY IS THE SOFTWARE AND NOT METACOGNITION. THE INTRODUCTION WAS MODIFIED TO BETTER PORTRAY THIS FOCUS, AND INTRODUCE CONCEPTS SUCH AS COMPLEXITY AND ORGANIZATION.

- [Consideration] There is greater opportunity to set up the problem you are attempting to solve. In this case, that concept maps are an effective tool to help student organize their knowledge but that they take significant resources to provide feedback.

CORRECT. THE INTRODUCTION WAS AUGMENTED TO REFLECT THIS OBSERVATION.

- [Consideration] Line 49 – As currently written, I could not identify any reflection or self-assessment activities in your concept map activity. Consider whether you are over-applying the theory of metacognition and self-regulation and whether it aligns to your study and learning activity.

AGREED. SOME OF THE REFERENCES TO REFLECTION/SELF-ASSESSMENT WERE REMOVED OR REFINED.

- [Recommendation] Can you greater specify your research aims? Investigate what? The utility, feasibility and usability of the program? You have compared the results of the software with PGY1 raters. How does this match to a more specific aim?

THE AIMS OF COMPARING AGAINST EXPERT GRADING AND MINIMIZING THE FEEDBACK PROCESS ARE MORE CLEARLY STATED.

Methods

- [Recommendation] Recommend to start section with a traditional population paragraph of what year the students were, how many cohorts, size of the cohort, degree program, country etc.

SOME DETAILS ADDED.

- [Recommendation] Include data collection procedures, how students were consented/informed about the study, recruitment procedures. Was this study approved by your ethics committee?

AGREED. DETAILS ADDED.

- [Recommendation] When I started to read the first paragraph of the Results Section, I was surprised to see these types of results. Could you set this up more in the Methods Section so the reader understands the data analysis? The methods section does do a great job of defining what nodes and links mean. However, it is less clear what aims you are meeting and why. It could be as simple as writing in the Methods Section that you calculated some descriptive characteristics of the students’ concepts maps by means of XYZ..etc.

SOME DETAILS ADDED, INCLUDING SUB-HEADINGS THAT SHOULD MAKE THE WRITE-UP MORE READABLE.

- [Recommendation] Similar to previous comment, can you map out chronologically your data analysis plan in the same order that is presented in the Results Section. Potentially, you might consider moving some of what is written in the Results Section to Methods.

SOME CONTENT WAS MOVED, AND THE FLOW WAS REVISED TO ADDRESS THIS RECOMMENDATION.

- [Consideration] Can you set up your reasoning for comparing the answers of the software to PGY1 raters. In this case, explain to the reader that this provides evidence of validity of the software program. You could cite similar studies to demonstrate that this is a typical process in software development? The readers would be novices in this subject area.

THIS IS AN EXCELLENT SUGGESTION AND IS NOW INCLUDED.

Results [Recommendation]

- p5. Line 138-142: You have provided a correlation before providing the raw data. What is the “complexity measure” and “organization metric” for the software? For the PGY1 raters? Please provide this data in text or table first before the correlation. Or maybe I am confused by the terms.

THE PARAGRAPH HAS BEEN REVISED TO BE MORE CLEAR.

- P5, line 143: What does a check on effort mean? How is the re-run different than the original analysis? It may be the sentence structure of the first few sentences that is unclear. So did you re-run everything after applying a spell checker? Or after a human looked for matches?

- line 155: What is the “simple text-replacement program”?

THIS PARAGRAPH HAS ALSO BEEN REVISED.

- Line 159 – What is the significance of a 22% and 24% overlap? Any expert opinion on how significant those values are? Also, what does this mean? What exactly is the difference between the PGY1 coded maps and the “original maps with text replacement”? Are you saying that PGY1s marked the students maps AND that they were the ones that re-coded for spelling errors? Again, I think this would all make more sense if the Methods Section followed a clear chronological order for what calculations and procedures were conducted. I recommend to be especially clear as to who (i.e., research team, PGY1s, students) or what (i.e., software, statistical programs) is doing each calculation/procedure.

THIS PARAGRAPH WAS HEAVILY REWRITTEN TO BE MORE CLEAR.

- If you have this available - how much time did it take the human graders?

THIS EFFORT WAS ESTIMATED BY THE TRANSCRIBERS AND IS NOW GIVEN IN THE RESULTS.

- Table 2: Does the N=8 mean that only 8 student concept maps were evaluated?

THIS IS CORRECT, EIGHT FOR EACH DISEASE STATE.

Discussion

- You have discussed complexity and organization but what about the accuracy of students’ maps? Was this evaluated? Will this ever be able to be assessed by the software OR will this component always need to be provided by a human rater?

ACCURACY WAS NOT STUDIED IN THIS WORK. A STATEMENT TO THAT EFFECT IS GIVEN IN THE LIMITATIONS.

- [Consideration] Authors could choose to improve clarity by using more active voice and less passive voice sentences. Line 206 is a perfect example of this.

- [Consideration] Consider adding a statement of whether this software is ready for use in a classroom.

DONE.

As I am still confused by some results, I cannot fully comment on the Discussion section. I am left thinking the software just provides a metric for the student concept map’s level of complexity and organization. I am unsure if it provides a % overlap with each map and the key. I am unsure if students can use the software for formative assessment – i.e., does it show students where they are lacking complexity or organization?

WITH THE SUBSTANTIVE CHANGES TO THE MANUSCRIPT IT IS HOPED THAT THE REVIEWER'S CONSFUSION IS LESSENED AND THEY ARE ABLE TO ANSWER THOSE QUESTIONS.

Round 2

Reviewer 1 Report

The authors certainly made a substantial improvement in the text. The methodology applied and the results are clearer and more understandable than before, although it is not yet an easy to understand text. However, the introduction continues to focus on megacognition and not on software to support the evaluation of didactic activities, or more specifically, concept maps.

Author Response

Reviewer: The authors certainly made a substantial improvement in the text. The methodology applied and the results are clearer and more understandable than before, although it is not yet an easy to understand text. However, the introduction continues to focus on megacognition and not on software to support the evaluation of didactic activities, or more specifically, concept maps.

We thank the reviewer for their comments. We have revisited the manuscript to try to be more consistent in our terminology and detailed in our methods and conclusions.

We agree with the reviewer that the paper starts with a description of metacognition. We have tried to "tone down" that focus. By the end of the first paragraph the writing has been linked to concept mapping. The second paragraph then further explains concept mapping. The third paragraph explains why this activity is important in pharmacy schools, and the fourth paragraph justifies our software.